# Exploring Social and Financial Hardship, Mental Health Problems and the Role of Social Support in Asylum Seekers Using Structural Equation Modelling

**DOI:** 10.3390/ijerph17196948

**Published:** 2020-09-23

**Authors:** Mathilde Sengoelge, Øivind Solberg, Alexander Nissen, Fredrik Saboonchi

**Affiliations:** 1Department of Health Sciences, Red Cross University College, 141 57 Huddinge, Sweden; oivind.fjeld-solberg@nkvts.no (Ø.S.); a.f.w.nissen@nkvts.no (A.N.); sabf@rkh.se (F.S.); 2Division for Implementation and Treatment Research, Norwegian Centre for Violence and Traumatic Stress Studies, 0484 Oslo, Norway; 3Division of Insurance Medicine, Department of Clinical Neuroscience, 171 77 Stockholm, Sweden

**Keywords:** asylum seekers, mental health, social and financial hardship, social support

## Abstract

Asylum seekers are exposed to a range of social and financial difficulties suggested to adversely impact mental health. Uprooted social networks and living conditions during the asylum seeking process potentially predispose this population to low access to social support. The aim of this study was to examine the relationship between social and financial hardship and mental health problems, and assess the potential mediating role of social support among asylum seekers. Cross sectional survey data from a cohort of asylum seekers in Sweden (N = 455) were subjected to structural equation modelling for examining hypothesized pathways between social and financial hardship, common mental health problems (CMHPs) and social support. Fit indices showed adequate to excellent fit of the examined models with CMHPs as the outcome (all CFI ≥ 0.951, RMSEA < 0.05, SRMR < 0.056). CMHPs were positively regressed on social and financial hardship (B = 0.786, S = 0.102, *p* < 0.001) and negatively regressed on social support (B = −0.103, SE = 0.032, *p* = 0.001). Social support mediated the association between social and financial hardship and CMHPs (effect estimate = 0.075, 95% CI = 0.032–0.136). The results point to the importance of social and living conditions of asylum seekers and indicate that social support is both socially patterned and may act as a mitigating resource to inform interventions and policies.

## 1. Introduction

At present approximately 500,000 asylum seekers are still awaiting decisions on granting refugee status or subsidiary protection in the EU [1] Asylum seekers are individuals who have not yet received the right to international protection in a host country. During the asylum procedure asylum seekers often receive accommodation on a no-choice basis, may experience detention, and anxiety and worry due to the uncertainty of the asylum outcome. [2] Moreover, the average application processing time in Europe ranges from several months to even years after lodging an application. [3]. In fact, the asylum seeking process in itself has been shown to be detrimental to applicants’ mental health in Europe and globally [4,5,6], by triggering, sustaining or even worsening existing mental health problems [7,8]. This relationship remains even after controlling for pre-migratory trauma predictors [9] and the levels of mental health problems are higher compared to refugees with formal refugee status [10,11].

In addition to the stress of the application process, asylum seekers also experience post-migratory social and financial hardship that adversely affect their mental health [12,13]. Social hardship refers to social exclusion as well as the loss of social status and networks, and perceived discrimination in the host society. The financial hardship is linked to the restrictions placed by host governments on employment and livelihood resulting in being deprived of basic needs and necessities. Among forced migrants, reoccurring daily stressors faced in the host country have been shown to be associated with the severity of common mental health symptoms [14], and may have an equal or even stronger relationship with psychological problems than exposure to pre-migratory traumatic events [15,16]. Our own research on asylum seekers in Sweden has suggested that asylum seekers’ psychosocial situation may in fact act as a psychosocial vulnerability factor whereby post-migratory stressors, such as “feeling disrespected due to national background” and “feeling excluded or isolated”, act as consistent correlates of depression and anxiety [17]. However, a limited focus on pre-migratory trauma as the main determinant of mental health of forced migrants may have contributed to obscuring the impact of concurrent social stressors anchored in refugee and asylum seekers living conditions [18,19]. 

Social support [20] has been shown to have an important explanatory role in the relationship between stressors and psychological health [21,22]. For asylum seekers and refugees, social support in the host country may act as a buffer against mental health problems by providing tangible and instrumental support in coping with stressors, as well as by offering a sense of belonging, which in turn may reduce the likelihood of psychological problems [23]. Social support from close social ties, for example living with one’s partner, has also been shown to potentially act as a protective factor for mental health among Syrian refugees who experienced torture [24]. Asylum seekers who are already uprooted from their social networks and living in temporary facilities are prone to being isolated, and may have little access to support resources, further exacerbating the impact of stressors in the host country. Given that social support may be influenced by asylum seekers’ adverse living conditions, and at the same time, has the potential to act as a buffer against experienced adversities, the relationship between social stressors, mental health and social support within the context of asylum seeking, warrants further investigation.

Consequently, the purpose of this study was to examine whether social and financial hardship assert an influence on common mental health problems (CMHPs) among asylum seekers, and further, to assess the potential mediating role of social support. Given the challenges associated with acquiring substantive longitudinal data among asylum seekers due to uncertainties surrounding residency and living conditions, a structural equation modelling (SEM) framework was used to evaluate the implied causal assumptions. Although SEM cannot establish causality on the basis of observational data, it provides an opportunity to examine the plausibility of statistically modelled effects among variables drawn from theory and previous research [25].

The examined model in the present study was based on the framework of social determinants of mental health that recognizes that mental health is shaped to a large extent by social, economic and environmental factors [18,26] and social support [24]. There is considerable evidence of the association between lower economic status and CMHPs, and these systematic differences in mental health are inequitable and require action on addressing the greater exposure and vulnerability to these social determinants [19]. The study has the following hypotheses: (1) Higher levels of social and financial hardship are associated with higher levels of CMHPs as well as with lower social support; (2) Social support is, in turn, negatively associated with CMHPS, mediating the relationship between social and financial hardship on CMHPs (see Appendix A for visualization of the hypothesized model).

## 2. Materials and Methods 

A cross sectional survey design was utilized to obtain data from asylum seekers residing in three large housing facilities from May 2016 to March 2018 in Sweden. Inclusion criteria for participation in the study included: status as asylum seeker from one of the largest asylum seeking groups in Sweden at the time of data collection (Afghanistan, Eritrea, Iraq, Somalia or Syria), and being at least 18 years of age. The Swedish ethics committee for research approved the data collection. A total of 455 asylum seekers returned a completed questionnaire, provided in the appropriate language to eligible participants (26.8% of all 1698 eligible housing residents). Of the 455 respondents, 4.5% arrived in Sweden prior to 2014, 86.9% arrived in 2014 or 2015 and 8.6% arrived in 2016–2018. Table 1 shows the characteristics of the sample.

### 2.1. Measures

#### 2.1.1. Social and Financial Hardship

Three subscales of the Refugee Post-Migration Stress Scale (RPMS) [14] were used to measure the construct of social and financial hardship based on asylum seekers’ responses to: (1) experiencing *perceived discrimination*; (2) *material and economic strain*; and (3) *social strain* while living in housing facilities in Sweden. RPMS is a concise, multidimensional self-assessment instrument for the evaluation of post-migration stress among refugees based on a theoretical conceptualization and empirical evidence of the construct of post-migration stress. Each subscale comprises of three to four items, all scored on a 5-point Likert scale ranging from 1 (*Never*) to 5 (*Very often*). A sample item for the “perceived discrimination” subscale was: Please indicate how frequently you experience the following situation: “Feeling disrespected due to my national background”. Cronbach’s alpha was 0.83 (*perceived discrimination*); 0.75 (*material and economic strain*); 0.81 (*social strain*). 

#### 2.1.2. Common Mental Health Problems (CMHPs) 

The Hopkins Symptom Checklist (HSCL-25) was used to measure the common mental health problems of anxiety and depression [27]. The HSCL-25 scale consists of 25 items; 10 items for anxiety symptoms and 15 items for depression symptoms. For all 25 items the response is rated on a 4-point Likert scale ranging from 1 (*Not at all*) to 4 (*Extremely*). A higher score indicates more symptom severity. It has been validated in refugee populations [28,29] and Cronbach’s alpha for the HSCL anxiety and depression subscales were 0.93 and 0.94, respectively. 

#### 2.1.3. Social Support

The ENRICHD Social Support Instrument (ESSI) was used to measure perceived social support. The ESSI is a seven-item self-report instrument that measures structural, instrumental and emotional support [20]. The first six items are scored on a 5-point Likert scale ranging from 1 (*None of the time*) to 5 (*All of the time*). The seventh item is a binary Yes/No question on cohabitation (Are you currently married or living with a partner?). The inventory has been validated in refugee populations [24]. Cronbach’s alpha for the first six items with the Likert-type response scale was 0.93. 

#### 2.1.4. Sociodemographic Variables 

Data on gender, age, country of origin and year of immigration were provided by the Swedish Migration Board. Age of respondents was categorized into two groups as 18–30 years and 31–64 years. Data on highest level of education were self-reported. 

### 2.2. Statistical Analysis 

Cross tabulations and simple summary statistics were used to make frequency and descriptive tables. Following the standardized scoring procedure for HSCL25 [27], mean-item score for the HSCL anxiety and depression sub-scales were calculated by summing all item scores in each sub-scales and dividing by the number of items answered. These scores were subsequently used for modelling CMHPs as a latent variable (see Section 2.2.2 Measurement Models). In order to receive a mean-item score, a respondent had to answer ≥23 items on the 25-item scale. 

#### 2.2.1. Structural Equation Modelling

As recommended for analyses with latent variables, a two-step modelling approach [30] was utilized, namely establishing the fit of the measurement model (step 1) and the fit of a model including the relationships between these latent variables, i.e., the full structural model (step 2). Establishing the fit of the measurement models is a prerequisite for determining that detection of potential misfits in the structural models are not due to inadequacy in measurements. The following goodness-of-fit indices were used to evaluate model fit: Satorra–Bentler scaled chi-square (S-Bχ^2^); comparative fit index (CFI) with cut-off values of ≥0.90 indicating adequate fit [31] and ≥0.95 indicating good fit; root mean square error of approximation (RMSEA) with 90% confidence intervals with RMSEA < 0.06 indicating good fit; and the standardized root mean square residual (SRMR) with SRMR < 0.08 indicating good fit [32]—Using several indices of fit reduces the chance of rejecting a well-fitted model based on a single index, specifically given the sensitivity of χ^2^ statistics to sample size and model complexity [33]. Modification indices (MIs) were examined to evaluate model re-specifications for potential fit improvements. Comparison between nested models was performed through a Satorra–Bentler chi-square difference likelihood ratio (ΔS-Bχ^2^) test [34] and a non-significant ΔS-Bχ^2^ indicates that a model with larger degrees of freedom does not worsen the model fit. Maximum likelihood estimation with robust standard errors (MLR) was used for all analyses.

#### 2.2.2. Measurement Models 

In accordance with the SEM approach to the analyses, all self-reported measures were treated as latent variables with the individual item responses modeled as reflective indicators [35]. With regards to social support, ESSI was modelled with the first six Likert-type items as reflective indicators (see Appendix A). On the basis of a previous model in refugee populations [24], a re-specification including estimation of a covariance between residuals for two items was assessed and compared by a ΔS-Bχ^2^ test). Given the content and response scale differences in the cohabitation item, this item was used as a formative indicator [35] of social support in all proceeding analyses.

*Social and financial hardship* was first modelled as three correlated first-order latent variables using the subscales of the RPMS *perceived discrimination* (four items), *material and economic strain* (four items), and *social strain* (three items) on the basis of the previously suggested factorial structure of the RPMS [14]. Next, a modeled second order “umbrella” latent variable of *social and financial hardship* was examined with these three latent variables as indicators (see Appendix A). This hierarchical modeling of the construct reflects the theoretical underpinning of the measurement. For the purpose of identification, at this stage the residual variance of *social strain* was fixed to zero. 

Finally, CMHPs were modelled as a latent variable with the separate scores of anxiety and depression driven by the standardized scoring procedures of HSCL25. Model identification for CMHPs at this stage, however, could not be achieved due to the variable being modelled with only two indicators [36]. This latent variable was therefore examined at the proceeding stage of the analyses within the full structural model where the model could be properly identified [33]. 

#### 2.2.3. Full Structural Model 

The main model included a structural model in which CMHPs were regressed on both social and financial hardship and social support. CMHPs were modelled as a latent variable with the HSCL anxiety and depression sub-scales as indicators. The model also included a link in which social support was regressed on social and financial hardship. This hypothesized pattern designates the function of social support as a mediator between social and financial hardship and CMHPs. The estimates of individual links and regression weights were inspected once model fit was established. In regard to the mediation hypothesis, the total statistically modelled effect of social and financial hardship on CMHPs was broken down into a direct effect and an indirect effect. The significance of the indirect/mediated effect via social support was assessed by examining the bias adjusted bootstrap confidence interval (1000 bootstraps) to account for the non-normal distribution of the mediation effect. The model also included gender and age as covariates. Cohabitation status was included as a formative indicator for social support in the model [24,35]. In order to assess the fit of the latent modelling of CMHPs that was not possible in step 1, this model was also compared to a model in which CMHPs were replaced with the correlated mean-item scores of HSCL anxiety and depression subscales by ΔS-Bχ^2^ statistics. For sensitivity analyses, an alternative model in which social and financial hardship was modelled as an outcome of CMHPs was compared to the main hypothesized model by means of Bayesian information criterion (BIC). For frequency and descriptive statistics, SPSS V24.0 was used, and all analyses with SEM were performed using the Mplus V8.3 software.

## 3. Results

Summary statistics for indicator variables used in the structural equation models are presented in Table 2.

### 3.1. SEM Analysis Step 1: Measurement Models

The fit indices for the measurement models of social support as measured by ESSI’s six items are displayed in Table 3. The unidimensional structure of ESSI displayed acceptable fit indices despite a significant S-Bχ^2^. The re-specified model allowing for free estimation of the error terms of two items resulted in significant enhancement of model fit indicated by a non-significant S-Bχ^2^ (10.89, df = 8, *p* = 0.208) and other excellent model fit indices. The re-specified model showed significant improvement of the fit for (ΔS-Bχ^2^ = 26.18, df = 1, *p* < 0.001) and is displayed in Figure 1.

Table 3 also contains the fit indices for the first order model of social and financial hardship. This first-order model produced acceptable values for model fit indices despite a significant S-Bχ^2^. Given the sensitivity of χ^2^ statistics to model complexity and sample size, the fit indices were deemed to indicate an adequate model fit. The second-order model also displayed acceptable fit indicated by highly acceptable fit indices, and ΔS-Bχ^2^ test indicated that this second-order model did not significantly worsen the model fit (ΔS-Bχ^2^ = 0.883, df = 1, *p* = 0.361). The second-order measurement model is displayed in Panel b of Figure 1. 

### 3.2. SEM Analysis Step Two: Full Structural Model

Upon establishing the adequacy of the measurement models, the fit of the full structural model with CMHPs as outcome was assessed. The fit indices for the full structural model also indicated adequate fit to the data by RMSEA = 0.048 (90% CI = 0.041–0.055), CFI = 0.951, and SRMR = 0.056, despite the expected significant S-Bχ^2^ (S = 383.62, df = 195, *p* < 0.001) due to model complexity indicated by the large DF and sample size. The ΔS-Bχ^2^ test for comparison between this model and the one in which the outcome of CMHPs was replaced by separate scores of anxiety and depression showed that modelling CMHPs as latent variables did not significantly worsen model fit (ΔS-Bχ^2^ = 2.636, df = 3, *p* = 0.451). The full structural model can be viewed in Figure 2.

According to this model CMHPs were significantly regressed on social and financial hardship, with higher scores on social and financial hardship associated with higher levels of CMHPs (B = 0.786, 95% CI = 0.586–0.986). CMHPs were also significantly and negatively regressed on social support (B = −0.103, 95% CI = −0.166 to −0.040), indicating that higher levels of social support were associated with lower levels of CMHPs. Finally, the model showed that social support, in turn, was negatively regressed on social and financial hardship, that is higher levels of social and financial hardship were associated with lower levels of social support (B = −0.726, 95% CI = −0.995 to −0.458). These significant associations are displayed with corresponding estimates in Figure 1. Gender showed significant associations with both CMHPs and social and financial hardship. Being female was associated with higher levels of CMHPs (B = 0.175, 95% CI = 0.038–0.312) but lower levels of social and financial hardship (B = −368, 95% CI = −0.554 to −0.182). The model accounted for a substantive significant proportion of the variance in CMHPs (R2 = 0.552, SE = 0.055, *p* < 0.001). Additionally, as can be seen in Table 4 the mediation analysis based on bias-corrected bootstrapped CIs showed that social support partially mediated the association between social and financial hardship and CMHPs (indirect effect estimate = 0.075, 95% CI = 0.032–0.136). The result of the sensitivity analysis indicated that the hypothesized model in the study fitted the data better (BIC = 22,029.08) than the alternative reverse model in which social and financial hardship was hypothesized to be influenced by CMHPs (BIC = 22,032.47).

## 4. Discussion

Findings from the present study add support to the growing recognition of social and financial hardship during the application process as important antecedents of mental ill health in asylum seekers. Given that the modelled influence of perceived discrimination, financial constraints, and social exclusion on CMHPs showed sufficient fit to the available data, our results corroborate the need for broadening the prevailing narrow focus on pre-migratory trauma [37]. The findings make an important empirical contribution to an emerging body of research demonstrating that asylum seekers’ mental health is influenced by the same social determinants as that of forced migrant populations [4,12,13] and of native born populations [18,19], such as poverty, inadequate housing, discrimination and social isolation (for a critical review see Hynie [18]). Moreover, psychosocial stressors related to the extended insecure status, perceived hostility and restrictions during the asylum seeking process may further aggravate the impact of adverse social and financial conditions [17]. Given the revealed substantive associations between social and financial hardship and CMHPs in the present study, our result could be interpreted as ascribing an important significance to social conditions ranging from the post-arrival to pre-resettlement.

A key finding of this study is that social support is a potential pathway that mediates the link between social and financial hardship and the mental health of asylum seekers. While there is ample evidence of the protective function of social support in both general populations [20] and forced migrants [24], the present finding on mediation contributes to an account of how social and financial adversities may also indirectly impact mental health by undermining access to social support, as well as hinder opportunities to mobilize this important protective resource during the asylum seeking process. Indeed, financial strain experienced by asylum seekers [38] has been suggested to adversely affect mental health by creating conditions in the host country that make accessing social support difficult. Similarly, perceived discrimination not only affects mental health negatively [12], but may also decrease social engagement [21,39]. Additionally, the experience of social exclusion and isolation related to asylum seekers being confined in housing facilities and lacking citizenship can be detrimental, not only to mental health [10], but also decrease the perceived levels of social support. As has been suggested [40], this implies that social support as a coping resource is both socially patterned and a potentially mitigating factor for the harmful indirect effect of stressors [21]. Our results concerning the mediating function of social support substantiate both of these notions among asylum seekers. Enhancing access to social support may, thus, be an important target of and consideration for interventions and policy decisions in efforts to protect and promote the mental health of asylum seekers.

In all, our results suggest the importance of a public health approach to address the mental health of asylum seekers by targeting social and financial drivers of health, as well as by improving access to, and quality of, social support resources [7,8,17]. Such an approach would require increased collaboration between policy makers, mental health services and refugee resettlement programs [41] and may be informed by further research on the social determinants of mental health in this forced migrant population. Understanding the role of these determinants is essential in order to implement effective interventions that increase protective factors such as family reunification for social support and allowing employment opportunities to ensure financial security. Such actions to promote mental well being are essential to improve the health of refugee populations as well as reduce health inequities.

This is one of the few studies based on the framework of social determinants of mental health applied in asylum seekers. However, our study has important limitations. Due to the cross-sectional design of the present study, causal directions in the links between social and financial hardship and CMHPs cannot be conclusively determined. The SEM approach enabled us to assess the plausibility of the direction of associations by statistically modelling these assumptions [25]. The results are an encouraging point of departure for future intervention studies and policy evaluations. Another limitation is that the model excluded other potential stressors related to the asylum seeking process (e.g., fear of being sent home and lack of adequate information), as well as pre-migratory trauma. Nonetheless, the model showed sufficient fit to the data, was preferable to an alternative model of reversed causality and accounted for a substantive proportion of the variance in the included outcome. 

A further limitation is that a convenience sampling method was used as asylum seekers upon arrival are unsystematically allocated to a few large housing facilities. The inevitable lack of random or representative sampling strategy may result in a number of sample specific differences that limit the generalizability of our findings. Finally, mental health self-report measures neither investigate other comorbid conditions that may produce similar symptoms, nor capture all aspects of mental health. Yet, given the established validity of HSCL-25 in refugee populations the utilization of this survey methodology enables a sense of privacy and provides essential important data. Similarly, objective indicators for socioeconomic status were, due to the cohort’s lack of formal residency, not available. Although the addition of objective indicators of social and economic status would have provided a more complete picture of the cohort’s social conditions, subjective self-reporting of these constructs has been shown to be consistently related to psychological functioning, even when controlling for objective indicators. Therefore, it appears to be a valid target for investigation.

## 5. Conclusions

Social and financial adversities anchored in asylum seekers living conditions in the host country appear to be important antecedents of mental health problems, as well as obstacles to access to and mobilization of social support. As both a potential stress buffering factor and psychological resource, social support is socially patterned and warrants promotion in host country policies and programs addressing well-being of asylum seekers. Future research and policy evaluation are needed to enable conclusive identification of key determinants of mental health in this context, and assess the efficiency of socially anchored measures to improve the mental health of this vulnerable population.

## Figures and Tables

**Figure 1 ijerph-17-06948-f001:**
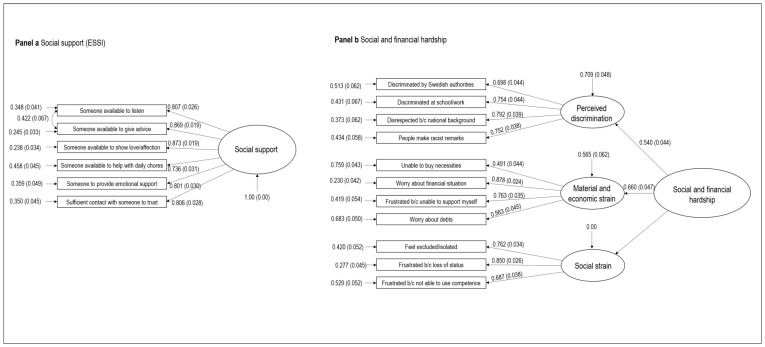
Measurement models in step one of Structural Equation Modeling procedure. Panel a shows final measurement model of social support; Panel b shows final measurement model for social and financial hardship. The displayed estimates are standardized coefficients (β) and standard error is displayed in parentheses.

**Figure 2 ijerph-17-06948-f002:**
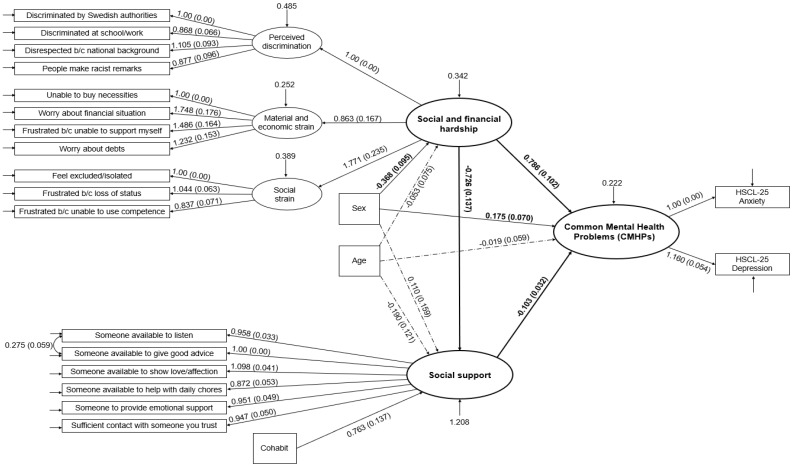
Full Structural Equation Model of Common Mental Health Problems (CMHPs) regressed on the latent variables social and financial hardship and social support. The displayed estimates for regression weights are unstandardized (B) with robust standard error in parentheses. Significant weights are indicated by solid-line arrows. Dashed-line arrows indicate nonsignificant regression weights. The weight estimates of main interest are in bold. Values for residual variances of indicator variables (boxes) are not included in the figure for readability issues (empty arrows signify that variances were part of the full structural equation model).

**Table 1 ijerph-17-06948-t001:** Sociodemographic characteristics of asylum seekers in Sweden (N = 455).

Characteristics	n	%
Gender		
Women	122	26.8
Men	333	73.2
Age groups		
18–30	269	59.1
31–64	186	40.9
Educational level		
<9 years	261	57.4
9–12 years	101	22.2
>12 years	74	16.3
Missing	19	4.2
Cohabitation/Family situation		
Living with a partner	119	26.2
Not living with a partner	261	57.4
Divorced/widow	37	8.1
Missing	38	8.4
Country of origin		
Afghanistan	154	33.8
Eritrea	45	9.9
Iraq	38	8.4
Somalia	64	14.1
Syria	145	31.9
Stateless	9	2.0

**Table 2 ijerph-17-06948-t002:** Descriptive statistics on indicator variables used in structural equation modelling (SEM) analyses. ESSI: ENRICHD Social Support Instrument.

	n	Mean	Variance	Min	Max
**Social and financial hardship ^1^**					
*Perceived discrimination*					
Discrimination by Swedish authorities	426	1.96	1.83	1	5
Discrimination in school or at work	403	1.62	1.19	1	5
Feeling disrespected due to my national background	413	1.93	1.80	1	5
People making racist remarks towards me	418	1.74	1.14	1	5
*Material and economic strain*					
Being unable to buy necessities	428	2.68	2.24	1	5
Worry about unstable financial situation	426	3.50	2.07	1	5
Frustration for not being able to support myself financially	423	3.52	2.21	1	5
Worry about debts	408	2.73	2.75	1	5
*Social strain*					
Feeling excluded or isolated in the Swedish society	421	3.04	2.54	1	5
Frustration due to loss of status in the Swedish society	418	2.73	2.45	1	5
Frustration because I am not able to make use of my competences in Sweden	422	3.48	2.29	1	5
**Social support (ESSI) ^2^**					
Is there someone available to whom you can count on to listen to you when you need to talk?	429	2.58	2.14	1	5
Is there someone available to you to give good advice about a problem?	427	2.44	2.05	1	5
Is there someone available to you who shows you love and affection?	419	2.84	2.44	1	5
Is there someone available to help with daily chores	421	2.28	2.11	1	5
Can you count on anyone to provide you with emotional support (talking over problems or helping you make a difficult decision)?	420	2.36	2.13	1	5
Do you have as much contact as you would like with someone you feel close to, someone in whom you can trust and confide in?	424	2.70	2.16	1	5
**Hopkins Symptom Checklist (HSCL-25) ^3^**					
Anxiety	395	2.07	0.80		
Depression	395	2.29	0.81		

^1^ All questions are scored on a 5-point Likert scale going from 1 (*Never*) to 5 (*Very often*). ^2^ All questions are scored on a 5-point Likert scale going from 1 (*None of the time*) to 5 (*All of the time*). ^3^ All questions in both subscales are scored on a 4-point Likert scale going from 1 (*Not at all*) to 4 (*Extremely*). The anxiety and depression sub-scales consist of 10 and 15 items, respectively.

**Table 3 ijerph-17-06948-t003:** Fit indices of measurement models.

Model	χ^2^	df	*p*	CFI	RMSEA (90% CI)	SRMR	ΔS-Bχ^2^	∆df	*p*
**Social support (ESSI):**									
Model A: Unidimensional	42.58	9	<0.001	0.967	0.092 (0.066–0.121)	0.024			
Model B: Respecified unidimensional ^a^	10.89	8	0.208	0.997	0.029 (0.00–0.067)	0.012			
Model B vs. A							26.18	1	<0.001
**Social and financial hardship:**									
Model A: First-order 3-factorial	89.71	41	<0.001	0.963	0.054 (0.039–0.069)	0.039			
Model B: Second-order 3-factorial ^b^	90.23	42	<0.001	0.964	0.053 (0.038–0.068)	0.037			
Model B vs. A							0.833	1	0.361

CFI Comparative Fit Index; RMSEA Root Mean Squared Error of Approximation; CI Confidence Intervals; SRMR Standardized Root Mean Square Residual; Δ S-Bχ^2^ Satorra–Bentler scaled chi Square difference; ∆df difference degrees of freedom. Selected models are in bold. ^a^ Model includes a covariance between error terms of 2 items. ^b^ Model includes residual variance of a first-order indicator constrained to 0 for identification.

**Table 4 ijerph-17-06948-t004:** Direct, indirect, and total effect estimates and bias-corrected (BC) bootstrapped 95% CI of social and financial hardship on common mental problems (CMHPs).

	Unstandardized Estimate	BC 95% CI
Direct effect:		
Social and financial hardship → CMHPs	0.786	0.598–1.021
Indirect effect:		
Social and financial hardship → social support → CMHPs	0.075	0.032–0.136
Total effect:		
Direct effect + Indirect effect	0.860	0.665–1.122

CI: Confidence interval based on 1000 bootstraps.

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
