# Peer review of "Exploring Social and Financial Hardship, Mental Health Problems and the Role of Social Support in Asylum Seekers Using Structural Equation Modelling"

_ijerph, 2020, doi:10.3390/ijerph17196948_

Round 1
Reviewer 1 Report
This paper aimed "to examine whether social and financial hardship assert an influence on common mental health problems among asylum seekers, and further, assess the potential mediating role of social support."
Suggestions and questions (answers can/should be used to improve the paper):
1. Figures 1, 1S and 2S are in low quality. It is impossible to see them.
2. Both figures and table of the supplementary material could be in the paper (not as supplementary material).
3. The following sentence should be rephrased: "The results are in line with an emerging body of research demonstrating that asylum seekers’ mental health is influenced by the same social determinants as that of the general population[18] [19], as well as by conditions specific to forced migrant populations[4,12,13], such as poverty, inadequate housing, discrimination and social isolation (for a critical review see Hynie[18])." What is "general population"? The sentence might sound like asylum seekers are not considered present in the "general population". But I think they compose today the "general population", right?
4. Consider the previous sentence and the discussion section, if results are in line with other different researches, what is the contribution of the paper? It apparently shows the same results. The contribution of the paper should be explicitly presented in the paper.
5. What are the limitations of the study? They should be acknowledged in the discussion section.
Author Response
Reviewer 1
1. Suggestions and questions (answers can/should be used to improve the paper):
Figures 1, 1S and 2S are in low quality. It is impossible to see them.
Author Response: Figures are now in high quality.
2. Both figures and table of the supplementary material could be in the paper (not as supplementary material).
Author Response: We have added Figure 2S and table 1S into the paper and the figure 1S on the hypothesized model we propose remains in the supplemental so that we have a total of 4 tables and 2 figures for readers to view in the paper. We hope that the reviewer finds the addition of this material sufficient.
3. The following sentence should be rephrased: "The results are in line with an emerging body of research demonstrating that asylum seekers’ mental health is influenced by the same social determinants as that of the general population[18] [19], as well as by conditions specific to forced migrant populations[4,12,13], such as poverty, inadequate housing, discrimination and social isolation (for a critical review see Hynie[18])." What is "general population"? The sentence might sound like asylum seekers are not considered present in the "general population". But I think they compose today the "general population", right?
Author Response: We have now reworded the sentence to improve clarification, and replaced ‘general population’ with residents. It now reads:
The results are in line with an emerging body of research demonstrating that asylum seekers’ mental health is influenced by the same social determinants as that of forced migrant populations[4,12,13] and of native born populations [18] [19], such as poverty, inadequate housing, discrimination and social isolation (for a critical review see Hynie[18]).
4. Consider the previous sentence and the discussion section, if results are in line with other different researches, what is the contribution of the paper? It apparently shows the same results. The contribution of the paper should be explicitly presented in the paper.
Author Response:Edits have been made to the discussion section lines 232, 241 to make explicit the contribution of this study, namely that social support is a potential pathway that mediates the link between social and financial hardship and the mental health of asylum seekers. Also as this approach to forced migrants’ mental health is a fairly new direction within a field which is still, to a large extent, focused on pre-migratory trauma, the present study provides much needed empirical evidence for corroborating a social determinant approach to asylum seekers mental health.
5. What are the limitations of the study? They should be acknowledged in the discussion section.
Author Response: We have added a sentence in the discussion stating ‘However, the study has important limitations.’ We then edited the text to elaborate and make more explicit the limitations of the study (lines 265, 269).
Reviewer 2 Report
This study examines the mediating role of social support in the relations between social/financial hardship and common mental health problems (CMHPs). The manuscript is well-written. The abstract and literature review are concise, and the logic and hypotheses are clearly presented in the literature review. However, there are still some issues to be addressed.
In Table 1, why is the frequency of years of immigration and country of origin missing? I would recommend adding them back to reflect consistency in data presentation.
In 2.1.1 measures for social and financial hardship, please state how many items were present in each subscales. State how the scores were computed. Did the authors take an average for each subscale? Or did they sum the score? Or, (as I read later in the manuscript) each item was entered into the structural model separately? Also, state at least one sample item for each subscale so that readers know what the scale looks like. Finally, state the rating scale/response options – were the subscales rated on a Likert scale? Yes/No? Please clarify.
This extends to all measures in the measure section, including CMHPs and Social Support. Please make sure all necessary components stated above are present. Reporting Cronbach’s alpha alone is not sufficient.
Again, because in the measurement, the rating scale was not presented, when readers get to Table 2, the mean and variance cannot be understood. The authors should add a table note in Table 2 to remind readers what the range of rating scale is for each item.
Also, Cronbach’s alpha has already been reported in section 2. It is not necessary to report it again in section 3.
In the model, I notice that there is a variable “cohabit”, but I could not find any mentioning of how this variable was measured in the measure section. Also, why cohabitation needs to be controlled in the model? Rationale and literature support should be provided. Also, why is it only controlling social support and social/financial hardship, but not CMHPs? Again, clarification is needed.
Why did the authors present p-values in some places (e.g., line 216 and line 218), but CI in other places (line 221, for example)? CI is always preferred, but consistency is important too.
In the discussion, the authors should also discuss the limitation of a cross-sectional/correlational mediation model. Although the authors suggested that the pathway goes from hardship to social relations to mental health; however, these are correlational data, so it’s possible that the direction goes another way (e.g., mental health problems could make participants perceive more hardship, resulting in lower social relations. OR, lower social relations could lead to participants experiencing more hardship, as no one may be available to help them, thus resulting in mental health problems). The authors need to make it clear that this is only a correlational findings, but not in any way infer causation.
Reviewer 3 Report
I enjoyed reading this manuscript of a timely study on an important topic. Please see the following comments/suggestions being made to strengthen the article.
- Introduction – the authors explain the value of social support for mental health status specific to the study population. However, this section does not discuss the importance of the other study variables, social hardship and financial hardship.
- I encourage the authors to provide a discussion of the current literature on the role of both social hardship and financial hardship as they are framed in the study: perceived discrimination, material and economic strain, and social strain.
- Also, on lines 74-75 the authors refer to the social determinants of health (SDOH) framework but don’t provide any further information on this. Providing information on this framework and the role of social support, social hardship, and financial hardship as social determinants of health would better ground the study in this framework. As currently presented, the study does not seem to be grounded in the SDOH framework.
- Materials and Methods – the study approach and population are described well, as are the measures. Explaining the language in which surveys were written would be helpful. Also, were all eligible housing residents given a questionnaire? If not, how many were given the questionnaire?
- Results – the results are presented clearly with supporting tables. I believe there is a typo in line 213 (p. 7) where the “. Q” is in front of “social support”
- Discussion – The discussion is clear and well explained. On the bottom of p. 8, I encourage the authors to consider other types of research that could contribute to strengthening public health approaches. Understanding the role of SDOH is valuable to delineating both risk and protective factors for physical and mental health status. However, in order to identify and implement public health approaches to improving health status, research on effective interventions that increase protective factors, such as social support and financial security, are essential.
A small point throughout is to be consistent when hyphens for phrases such as “asylum seekers.” “pre-migration” and “post-migration.”
Author Response
Reviewer 3
I enjoyed reading this manuscript of a timely study on an important topic. Please see the following comments/suggestions being made to strengthen the article.
1. Introduction – the authors explain the value of social support for mental health status specific to the study population. However, this section does not discuss the importance of the other study variables, social hardship and financial hardship. I encourage the authors to provide a discussion of the current literature on the role of both social hardship and financial hardship as they are framed in the study: perceived discrimination, material and economic strain, and social strain.
Author Response: We have added text on this to explain these terms more clearly in the introduction (lines 43-47).
Also, on lines 74-75 the authors refer to the social determinants of health (SDOH) framework but don’t provide any further information on this. Providing information on this framework and the role of social support, social hardship, and financial hardship as social determinants of health would better ground the study in this framework. As currently presented, the study does not seem to be grounded in the SDOH framework.
Author Response: Additional text and references on the SDOH framework have been added (lines 80-81) to now read: The examined model in the present study was based on the framework of social determinants of mental health that recognizes that mental health is shaped to a large extent by social, economic and environmental factors [18,26] and social support [24]. There is considerable evidence of the association between lower economic status and CMHPs and these systematic differences in mental health are inequitable and require action on addressing the greater exposure and vulnerability to these social determinants[27].
2. Materials and Methods – the study approach and population are described well, as are the measures. Explaining the language in which surveys were written would be helpful. Also, were all eligible housing residents given a questionnaire? If not, how many were given the questionnaire?
Author Response: A total of 1698 eligible housing residents were given a questionnaire and the survey and informed consent form was made available in the appropriate language to eligible participants (e.g. Arabic for Syrians). This information has been added to the manuscript line 87 to 88.
3. Results – the results are presented clearly with supporting tables. I believe there is a typo in line 213 (p. 7) where the “. Q” is in front of “social support”
Author Response: The Q was a typo and has been removed.
4. Discussion – The discussion is clear and well explained. On the bottom of p. 8, I encourage the authors to consider other types of research that could contribute to strengthening public health approaches. Understanding the role of SDOH is valuable to delineating both risk and protective factors for physical and mental health status. However, in order to identify and implement public health approaches to improving health status, research on effective interventions that increase protective factors, such as social support and financial security, are essential.
Author Response: The following text has been added to the discussion to emphasize how the role of the SDOH is valuable in interpreting the results (lines 294 to 298): Understanding the role of these determinants is essential in order to implement effective interventions that increase protective factors such as family reunification for social support and allowing employment opportunities to ensure financial security. Such actions to promote mental well being are essential to improve the health of refugee populations as well as reduce health inequities.
A small point throughout is to be consistent when hyphens for phrases such as “asylum seekers.” “pre-migration” and “post-migration.”
Author Response: This has been corrected throughout the manuscript.
Round 2
Reviewer 1 Report
The authors answered my questions. My concerns were solved.